# Risk of hospitalized and non-hospitalized gastrointestinal bleeding in ALLHAT trial participants receiving diuretic, ACE-inhibitor, or calcium-channel blocker

**Xianglin L. Du**[1]*, **Lara M. Simpson**[2], **Brian C. Tandy**[2], **Judith L. Bettencourt**[2], **Barry R. Davis**[2]

**1** Department of Epidemiology, Human Genetics and Environmental Sciences, School of Public Health, The University of Texas Health Science Center at Houston, Houston, TX, United States of America,
**2** Coordinating Center for Clinical Trials, Department of Biostatistics and Data Science, School of Public Health, The University of Texas Health Science Center at Houston, Houston, TX, United States of America

* Xianglin.L.Du@uth.tmc.edu

**Data Availability Statement:** The ALLHAT data, Medicare claims data, Medicare Part-D and National Death Index (NDI) data are not public-use

## Abstract

### Objectives

This post-trial data linkage analysis was to utilize the data of Antihypertensive and Lipid-Lowering Treatment to Prevent Heart Attack Trial (ALLHAT) participants linked with their Medicare data to examine the risk of hospitalized and non-hospitalized gastrointestinal (GI) bleeding associated with antihypertensives.

### Settings

ALLHAT was a multicenter, randomized, double-blind, active-controlled trial conducted in a total of 42,418 participants aged ≥55 years with hypertension in 623 North American centers. Data for ALLHAT participants who were aged at ≥65 have been linked with their Medicare claims data.

### Participants

A total of 16,676 patients (4,480 for lisinopril, 4,537 for amlodipine, and 7,659 for chlorthalidone) with complete Medicare claims data were available for the final analysis.

### Results

The cumulative incidences through March 31, 2002 of hospitalized GI bleeding were 5.4%, 5.8% and 5.4% for amlodipine, lisinopril, and chlorthalidone arms, respectively, but were not statistically significant among the 3 arms after adjusting for confounders in Cox regression models. The cumulative incidences of non-hospitalized GI bleeding were also similar across the 3 arms (12.0%, 12.2% and 12.0% for amlodipine, lisinopril, and chlorthalidone, respectively). The increased risk of GI bleeding by age was statistically significant after adjusting for confounders (HR = 1.04 per year, 95% CI: 1.03–1.05). Smokers also had a significantly

datasets. These are third party data. All researchers would be able to access these data in the following same manner as the authors: researchers may request the ALLHAT data with the approval from the ALLHAT Coordinating Center in Houston, the Medicare claims data and Medicare Part-D data with the approval from the Center for Medicare and Medicaid Services (CMS), and the National Death Index (NDI) data with the approval from the National Center for Health Statistics (NCHS). We confirm that the authors did not have any special access privileges that others would not have. We plan to share the statistical models and statistical programs that we used to analyze these data upon request. Here are statistical models on the Kaplan-Meier estimates and Cox regressions using STATA software: 1). Kaplan Meier estimates on the cumulative incidence of GI bleeding: sts list, by (Group) failure at(0 1 2 3 4 5 6) 2). Cox regressions on the hazard ratio of GI bleeding: stset Outcomeyear if Outcome <., failure(Outcome) stcox GroupComparison.

**Funding:** This study was supported by the National Institutes of Health (NIH) grant Number R01AG058971 (PI: Dr. Du; Co-I's: Dr. Davis and Dr. Simpson). The funders had no role in study design, data collection and analysis, decision to publish, or preparation of the manuscript.

**Competing interests:** The authors have declared that no competing interests exist.

higher risk of having hospitalized GI bleeding (1.45, 1.19–1.76). Hispanics, those who used aspirin or atenolol in-trial, had diabetes, more education, and a history of stroke had a significantly lower risk of having GI bleeding than their counterparts. Other factors such as gender, history of CHD, prior antihypertensive use, use of estrogen in women, and obesity did not have significant effects on the risk of GI bleeding.

## Conclusion

There were no statistically significant differences on the risk of hospitalized or non-hospitalized GI bleeding among the 3 ALLHAT trial arms (amlodipine, lisinopril, and chlorthalidone) during the entire in-trial follow-up.

## Introduction

Calcium channel blockers (CCB)s have been well documented to be efficacious in treating patients with hypertension [1–10]. However, there were some concerns about potential risk of gastrointestinal (GI) bleeding associated with the use of CCBs in treating patients with hypertension in a prospective cohort study [11] and in three other observational studies [12–14], whereas other case-control and retrospective cohort studies found no significant association between CCB and risk of GI bleeding [15–19]. A large clinical trial, Antihypertensive and Lipid-Lowering Treatment to Prevent Heart Attack Trial (ALLHAT), did not find a significant association between CCB (amlodipine) or angiotensin-converting enzyme (ACE) inhibitor (lisinopril) versus thiazide diuretic (chlorthalidone) and the risk of GI bleeding [20]. A later ALLHAT report of the *post-hoc* comparison of amlodipine with ACE inhibitor lisinopril found a significantly lower risk of GI bleeding in those receiving CCB [21]. A more recent ALLHAT study specifically focusing on the risk of hospitalized GI bleeding in association with various antihypertensive drugs concluded that hypertensive patients on amlodipine did not have an increased risk of GI bleeding compared to those in chlorthalidone or lisinopril arms [22]. Because previous reports were based on patients' Medicare or Veteran Affairs (VA) data up to September 24, 2001 while the ALLHAT's last in-trial follow-up data was completed on March 31, 2002, the estimate for the risk of GI bleeding would be more accurate if their last follow-up for capturing GI bleeding from Medicare claims was also completed on March 31, 2002. We have now linked the data of ALLHAT participants with their Medicare data through the entire in-trial period, and hence the data enabled us to both update the analysis of hospitalized GI bleeding to the end of in-trial follow-up and examine the risk of GI bleeding. Moreover, we also examined the risk of non-hospitalized GI bleeding associated with antihypertensive drugs, which has never been reported before among ALLHAT trial participants. This is critically important because not all patients with GI bleeding were to be hospitalized, and therefore there was a concern about underreporting of overall GI bleeding based on hospitalizations alone. Hence, the findings of this study should have high public health and clinical significance with respect to GI bleeding and routine intake of antihypertensive drugs.

## Methods

### Study population and data sources

The detailed methods of ALLHAT have been reported previously [20–22]. In brief, ALLHAT was a multicenter, randomized, double-blind, active-controlled trial conducted in a total of

42,418 participants aged 55 years or older with hypertension and at least 1 other coronary heart disease (CHD) risk factor in 623 North American centers. Those patients who were eligible and agreed to participate were randomly assigned to 4 treatment arms: an angiotensin-converting enzyme (ACE) inhibitor (lisinopril) (n = 9,054), calcium channel blocker (amlodipine) (n = 9,048), α-blocker (doxazosin) (n = 9,061), or a thiazide-type diuretic (chlorthalidone) (n = 15,255). The primary outcome was the incidence of combined fatal CHD or nonfatal myocardial infarction. Secondary outcomes were all-cause mortality, combined CHD-specific mortality and combined cardiovascular disease (CVD) (combined CHD, stroke, and heart failure [HF]) mortality. This trial's recruitment period was from February 1, 1994 to January 31, 1998 with the last date of active in-trial follow-up on March 31, 2002. Average follow-up was 4.9 years (ranging from 4 to 8 years) in all arms except for doxazosin-chlorthalidone comparison which was terminated early with a mean follow-up of 3.2 years due to a higher incidence of CVD events. Therefore, this study did not include those patients on doxazosin. We recently obtained the ALLHAT-Medicare linked data in order to study the long-term benefits and harms of antihypertensive drugs, in which our secondary outcome is to examine short and long-term side effects on GI bleeding associated with the use of antihypertensive drugs. Due to potential changing patterns of antihypertensive drug uses after the trial ended to the present time, this report only focused on the short-term side effect of GI bleeding through the end of in-trial period. Data for ALLHAT participants who were aged 65 or older have been linked with their Medicare claims data for the entire in-trial period from January 1, 1994 to March 31, 2002. Of a total of 33,357 participants (9,054 for lisinopril, 9,048 for amlodipine, and 15,255 for chlorthalidone), 553 participants were excluded due to randomization in Canada, 11,960 participants were excluded because of not being eligible for Medicare at their entry to the trial, and 4,168 participants were excluded because of their randomization in the Veteran Affairs (VA) system to which we did not have access to long-term follow-up, leaving 16,676 patients (4,480 for lisinopril, 4,537 for amlodipine, and 7,659 for chlorthalidone) with complete Medicare claims data in the final analysis for this study.

## Patient and public involvement statement

Patients or the public were not involved in the design, or conduct, or reporting, or dissemination plans of our research.

## Ethics statement

This study was to use the existing and de-identified ALLHAT-Medicare linked datasets and there was no patient contact, therefore the form of consent was not obtained. There is no health risk to the subjects under study. This post-trial data linkage analysis was approved by the Committee for Protection of Human Subjects at the University of Texas Health Science Center in Houston (Study ID: HSC-SPH-17-1035).

## Study variables, main exposure and outcomes

Main exposure for the study were antihypertensive drugs (lisinopril, amlodipine, and chlorthalidone) which were allocated to participants through randomization. Main outcomes of this study included the occurrence of hospitalized and non-hospitalized GI bleeding. Because the information on hospitalized and non-hospitalized GI bleeding as secondary outcomes was not collected in ALLHAT centers or clinics, we obtained the linked Medicare claims data from the Center for Medicare and Medicaid Services (CMS) for the ALLHAT participants. We searched Medicare claims through March 31, 2002 as the last date of in-trial follow-up in order to ascertain the risk of GI bleeding during the entire in-trial period. The occurrence of hospitalized GI

bleeding was identified through the inpatient hospitalization files using ICD-9 or ICD-10 codes (S1 Table). The occurrence of non-hospitalized GI bleeding was identified through the outpatient files or from physician office visit files using the ICD-9 or ICD-10 codes. These outcomes were compared among subjects in 3 arms who received lisinopril, amlodipine, or chlorthalidone.

ALLHAT baseline demographic and clinical data were incorporated into analyses, including age, race, ethnicity, gender, prior receipt of antihypertensive drug therapy, blood pressure (BP), body mass index, history of coronary heart disease, aspirin and estrogen use, cigarette smoking, history of atherosclerotic cardiovascular disease, history of myocardial infarction or stroke, history of coronary revascularization, history of other atherosclerotic cardiovascular disease, ST-T wave, HDL-cholesterol <35 mg/dL, LVH (left ventricular hypertrophy) by ECG or echocardiography, LVH by Minnesota code, and diabetes.

## Statistical analysis

Baseline characteristics among the study comparison groups were compared using chi-square statistics. Cumulative incidence (probability) of GI bleeding were calculated from the date of initial randomization to the end of in-trial follow-up (3/31/2002). Cumulative incidence of GI bleeding was calculated using Kaplan-Meier method and presented in number of GI bleeding cases per 100 persons and per 1,000 persons. In addition, Cox regression models were used to perform the time to event analysis to determine the risk of developing GI bleeding by the 3 in-trial antihypertensive drugs while adjusting for confounding factors. The proportionality assumption was assessed by checking whether the log-log Kaplan-Meier curves are parallel and do not intersect and also by adding an interaction term between antihypertensive medication and time variables to the Cox models. In these models, the interactions between the 3 in-trial antihypertensive drugs and other factors (e.g., age, gender, race, aspirin use, and smoking status) on the risk of GI bleeding were tested and the results with p values were presented.

## Results

Table 1 presents the comparison of baseline characteristics among the 3 trial arms (lisinopril, amlodipine, and chlorthalidone). Of a total of 16,676 subjects who were eligible by the end of in-trial follow-up on 3/31/2002, the baseline characteristics such as age, gender, race/ethnicity, prior receipt of antihypertensive drug therapy, blood pressure, history of coronary heart disease, aspirin and estrogen use, cigarette smoking, history of atherosclerotic cardiovascular disease, history of myocardial infarction or stroke, history of coronary revascularization, history of other atherosclerotic cardiovascular disease, left ventricular hypertrophy, diabetes and obesity are generally similar among the 3 trial arms (lisinopril, amlodipine, and chlorthalidone).

Table 2 presents the cumulative incidence of hospitalized GI bleeding, non-hospitalized GI bleeding, and combined all GI bleeding (hospitalized or non-hospitalized GI bleeding) over the entire in-trial follow-up period from February 1, 1994 to March 31, 2002, by 3 RCT arms. The cumulative incidence of hospitalized GI bleeding was 5.4%, 5.8% and 5.4% for amlodipine, lisinopril, and chlorthalidone arms, respectively. Although the cumulative incidence of GI bleeding was slightly lower in patients with amlodipine as compared to those with lisinopril, it was not statistically significant after adjusting for measured confounders in the time to event Cox regression models (Table 3). The cumulative incidence of non-hospitalized GI bleeding was higher than that of hospitalized GI bleeding, but was similar across the 3 arms (12.0%, 12.2% and 12.0% for amlodipine, lisinopril, and chlorthalidone arms, respectively). The cumulative incidence of combined all GI bleeding (hospitalized or non-hospitalized GI bleeding) was also similar across the 3 arms (13.7%, 14.4% and 14.0% for amlodipine, lisinopril, and

**Table 1. Baseline characteristics of participants by randomized treatment arms.**

| Characteristics | Participants, No (%) | | | | P value |
|---|---|---|---|---|---|
| | Chlorthalidone | Amlodipine | Lisinopril | Total | |
| Eligible number of participants | 7,659 | 4,537 | 4,480 | 16,676 | |
| Age, mean [range], years | 71.3 [55–110] | 71.3 [55–101] | 71.5 [55–98] | 71.4 [55–110] | 0.467 |
| 55–64 | 656 (8.6) | 408 (9.0) | 374 (8.3) | 1438 (8.6) | 0.854 |
| 65–69 | 2656 (34.7) | 1571 (34.6) | 1549 (34.6) | 5776 (34.6) | |
| 70 or older | 4347 (56.8) | 2558 (56.4) | 2557 (57.1) | 9462 (56.7) | |
| Sex | | | | | |
| Women | 4383 (57.2) | 2581 (56.9) | 2489 (55.6) | 9453 (56.7) | 0.191 |
| Men | 3276 (42.8) | 1956 (43.1) | 1991 (44.4) | 7223 (43.3) | |
| Race/ethnicity | | | | | |
| Black | 2694 (35.2) | 1615 (35.6) | 1560 (34.8) | 5869 (35.2) | 0.742 |
| Non-Black | 4965 (64.8) | 2922 (64.4) | 2920 (65.2) | 10807 (64.8) | |
| Hispanic ethnicity | | | | | |
| Hispanic | 1653 (21.7) | 965 (21.4) | 1006 (22.6) | 3624 (21.8) | 0.367 |
| Non-Hispanic | 5966 (78.3) | 3545 (78.6) | 3451 (77.4) | 12962 (78.2) | |
| Education, mean (SD), years* | 10.3 (4.2) | 10.4 (4.2) | 10.3 (4.3) | 10.3 (4.2) | 0.042 |
| Antihypertensive treatment | | | | | |
| Treated (prior to baseline) | 6926 (90.4) | 4128 (91.0) | 4089 (91.3) | 15143 (90.8) | 0.278 |
| Untreated | 732 (9.6) | 409 (9.0) | 391 (8.7) | 1532 (9.2) | |
| Aspirin use at baseline* | | | | | |
| Yes aspirin used | 2694 (35.7) | 1594 (35.6) | 1624 (36.8) | 5912 (36.0) | 0.428 |
| Women taking estrogen* | 599 (13.9) | 340 (13.4) | 322 (13.2) | 1261 (13.6) | 0.649 |
| HDL, mean (SD), mg/dl | 48.2 (15.0) | 48.6 (15.1) | 47.9 (14.7) | 48.2 (14.9) | 0.262 |
| HDL <35 mg/dl | 776 (10.1) | 490 (10.8) | 465 (10.4) | 1731 (10.4) | 0.505 |
| Diabetes classification* | | | | | |
| Diabetic | 3045 (43.1) | 1827 (43.7) | 1788 (43.3) | 6660 (43.3) | 0.837 |
| Non-diabetic | 4020 (56.9) | 2356 (56.3) | 2342 (56.7) | 8718 (56.7) | |
| Cigarette smoking | | | | | |
| Smoker (current) | 1283 (16.8) | 772 (17.0) | 722 (16.1) | 2777 (16.7) | 0.496 |
| Nonsmoker (non/former) | 6376 (83.2) | 3765 (83.0) | 3757 (83.9) | 13898 (83.3) | |
| History of CHD* | 2105 (27.7) | 1180 (26.2) | 1208 (27.2) | 4493 (27.1) | 0.211 |
| Atherosclerotic CVD (yes if any of 4 below) | 4356 (56.9) | 2471 (54.5) | 2549 (56.9) | 9376 (56.2) | 0.020 |
| History MI or stroke | 1931 (25.2) | 1124 (24.8) | 1081 (24.1) | 4136 (24.8) | 0.411 |
| History coronary revascularization | 1085 (14.2) | 567 (12.5) | 663 (14.8) | 2315 (13.9) | 0.004 |
| Other atherosclerotic CVD | 2091 (27.3) | 1213 (26.7) | 1246 (27.8) | 4550 (27.3) | 0.327 |
| ST-T wave | 809 (10.6) | 456 (10.1) | 480 (10.8) | 1745 (10.5) | 0.549 |
| LVH by Minnesota code | 348 (5.4) | 217 (5.7) | 210 (5.6) | 775 (5.5) | 0.805 |
| SBP, mean (SD), mmHg | 146.4 (13.0) | 146.3 (13.1) | 146.7 (12.8) | 146.5 (13.0) | 0.523 |
| DBP, mean (SD), mmHg | 82.8 (9.0) | 82.7 (9.1) | 82.6 (9.1) | 82.7 (9.0) | 0.855 |
| BMI, mean (SD), mg/kg$^2$ | 29.1 (6.1) | 29.1 (6.0) | 29.3 (6.1) | 29.2 (6.1) | 0.270 |
| Obesity | 2897 (37.9) | 1718 (38.0) | 1740 (39.0) | 6355 (38.2) | 0.492 |
| Lipid trial participants | 1851 (24.2) | 1072 (23.6) | 1028 (22.9) | 3951 (23.7) | 0.309 |

*Reduced denominator Chlorthalidone/Amlodipine/Lisinopril available for Aspirin: C = 7543/A = 4482/L = 4419; BMI: C = 7636/A = 4519/L = 4464; Diabetes: C = 7065/A = 4183/L = 4419; Education: C = 7053/A = 4172/L = 4112; Estrogen: C = 4296/A = 2541/L = 2439; HDL: C = 7265/A = 4277/L = 4224; History of CHD: C = 7612/A = 4507/L = 4449.

**Table 2. Cumulative incidence of hospitalized GI bleeding, non-hospitalized GI bleeding, and combined all GI bleeding in-trial from 1994 through 3/31/2002.**

| | Hospitalized GI Bleeding | | Non-hospitalized GI Bleeding | | Hospitalized or non-hospitalized GI Bleeding | |
|---|---|---|---|---|---|---|
| | n/N | Rate (%) | n/N | Rate (%) | n/N | Rate (%) |
| **Total** | 914 / 16676 | 5.5 | 2012 / 16676 | 12.1 | 2335 / 16676 | 14.0 |
| Antihypertensive RZ Group | | | | | | |
| Chlorthalidone | 411 / 7659 | 5.4 | 920 / 7659 | 12.0 | 1069 / 7659 | 14.0 |
| Amlodipine | 244 / 4537 | 5.4 | 544 / 4537 | 12.0 | 620 / 4537 | 13.7 |
| Lisinopril | 259 / 4480 | 5.8 | 548 / 4480 | 12.2 | 646 / 4480 | 14.4 |
| Age groups | | | | | | |
| 55–64 | 74 / 1438 | 5.1 | 165 / 1438 | 11.5 | 192 / 1438 | 13.4 |
| 65–69 | 232 / 5776 | 4.0 | 650 / 5776 | 11.3 | 724 / 5776 | 12.5 |
| 70 and older | 608 / 9462 | 6.4 | 1197 / 9462 | 12.7 | 1419 / 9462 | 15.0 |
| Gender | | | | | | |
| Female | 522 / 9453 | 5.5 | 1146 / 9453 | 12.1 | 1336 / 9453 | 14.1 |
| Male | 392 / 7223 | 5.4 | 866 / 7223 | 12.0 | 999 / 7223 | 13.8 |
| Race/ethnicity | | | | | | |
| Black | 357 / 5869 | 6.1 | 755 / 5869 | 12.9 | 885 / 5869 | 15.1 |
| Non-Black | 557 / 10807 | 5.2 | 1257 / 10807 | 11.6 | 1450 / 10807 | 13.4 |
| Hispanic ethnicity | | | | | | |
| Hispanic | 150 / 3624 | 4.1 | 451 / 3624 | 12.4 | 495 / 3624 | 13.7 |
| Non-Hispanic | 757 / 12962 | 5.8 | 1551 / 12962 | 12.0 | 1824 / 12962 | 14.1 |
| Antihypertensive treatment | | | | | | |
| Treated (prior to baseline) | 847 / 15143 | 5.6 | 1842 / 15143 | 12.2 | 2147 / 15143 | 14.2 |
| Untreated | 67 / 1532 | 4.4 | 170 / 1532 | 11.1 | 188 / 1532 | 12.3 |
| Aspirin use at baseline | | | | | | |
| Yes aspirin used | 314 / 5912 | 5.3 | 702 / 5912 | 11.9 | 820 / 5912 | 13.9 |
| No aspirin | 590 / 10532 | 5.6 | 1290 / 10532 | 12.2 | 1491 / 10532 | 14.2 |
| Women taking estrogen | | | | | | |
| Yes | 52 / 1261 | 4.1 | 159 / 1261 | 12.6 | 175 / 1261 | 13.9 |
| No | 459 / 8015 | 5.7 | 962 / 8015 | 12.0 | 1132 / 8015 | 14.1 |
| HDL cholesterol <35 mg/dl | | | | | | |
| Yes | 88 / 1731 | 5.1 | 223 / 1731 | 12.9 | 254 / 1731 | 14.7 |
| No | 826 / 14945 | 5.5 | 1789 / 14945 | 12.0 | 2081 / 14945 | 13.9 |
| Diabetes classification | | | | | | |
| Diabetic | 428 / 6660 | 6.4 | 842 / 6660 | 12.6 | 992 / 6660 | 14.9 |
| Non-diabetic | 409 / 8718 | 4.7 | 1023 / 8718 | 11.7 | 1166 / 8718 | 13.4 |
| Cigarette smoking | | | | | | |
| Smoker (current) | 162 / 2777 | 5.8 | 300 / 2777 | 10.8 | 357 / 2777 | 12.9 |
| Nonsmoker (non/former) | 752 / 13898 | 5.4 | 1712 / 13898 | 12.3 | 1978 / 13898 | 14.2 |
| History of CHD | | | | | | |
| Yes | 254 / 4493 | 5.7 | 548 / 4493 | 12.2 | 640 / 4493 | 14.2 |
| No | 649 / 12075 | 5.4 | 1447 / 12075 | 12.0 | 1672 / 12075 | 13.8 |
| Atherosclerotic CVD | | | | | | |
| Yes | 546 / 9376 | 5.8 | 1166 / 9376 | 12.4 | 1362 / 9376 | 14.5 |
| No | 368 / 7300 | 5.0 | 846 / 7300 | 11.6 | 973 / 7300 | 13.3 |
| History MI or stroke | | | | | | |
| Yes | 273 / 4136 | 6.6 | 512 / 4136 | 12.4 | 618 / 4136 | 14.9 |
| No | 641 / 12540 | 5.1 | 1500 / 12540 | 12.0 | 1717 / 12540 | 13.7 |

*(Continued)*

**Table 2.** (Continued)

| | Hospitalized GI Bleeding | | Non-hospitalized GI Bleeding | | Hospitalized or non-hospitalized GI Bleeding | |
|---|---|---|---|---|---|---|
| | n/N | Rate (%) | n/N | Rate (%) | n/N | Rate (%) |
| History of coronary revascularization | | | | | | |
| Yes | 133 / 2315 | 5.7 | 292 / 2315 | 12.6 | 337 / 2315 | 14.6 |
| No | 781 / 14361 | 5.4 | 1720 / 14361 | 12.0 | 1998 / 14361 | 13.9 |
| Other atherosclerotic CVD | | | | | | |
| Yes | 256 / 4550 | 5.6 | 576 / 4550 | 12.7 | 670 / 4550 | 14.7 |
| No | 658 / 12126 | 5.4 | 1436 / 12126 | 11.8 | 1665 / 12126 | 13.7 |
| Major ST segment depression | | | | | | |
| Yes | 98 / 1745 | 5.6 | 208 / 1745 | 11.9 | 236 / 1745 | 13.5 |
| No | 806 / 14800 | 5.4 | 1784 / 14800 | 12.1 | 2074 / 14800 | 14.0 |
| LVH by Minnesota Code | | | | | | |
| Hard LVH | 59 / 775 | 7.6 | 83 / 775 | 10.7 | 108 / 775 | 13.9 |
| No LVH | 701 / 13239 | 5.3 | 1580 / 13239 | 11.9 | 1830 / 13239 | 13.8 |
| Lipid trial participants | | | | | | |
| Yes | 179 / 3951 | 4.5 | 479 / 3951 | 12.1 | 537 / 3951 | 13.6 |
| No | 735 / 12725 | 5.8 | 1533 / 12725 | 12.0 | 1798 / 12725 | 14.1 |

chlorthalidone arms, respectively) (Table 2) and was not statistically significantly different among the 3 groups after adjusting for confounders in multiple Cox regression models (Table 3). For example, the hazard ratios of having GI bleeding in those receiving chlorthalidone and lisinopril were 1.05 (95% CI: 0.95–1.16) and 1.01 (0.91–1.13) respectively as compared to those receiving amlodipine, whereas the hazard ratio of GI bleeding was 0.97 (0.88–1.07) in those receiving lisinopril as compared to subjects receiving chlorthalidone (Table 3).

Table 2 also presents the cumulative incidence of GI bleeding by age, gender, race/ethnicity, previous use of antihypertensives, use of aspirin and estrogen, HDL cholesterol level, smoking status, and history of comorbid conditions (diabetes, CHD, CVD, MI, stroke, or CABG). For example, the cumulative incidence of hospitalized GI bleeding was higher in patients aged 70 or older (6.4%) than those 55–64 (5.1%) or 65–69 (4.0%). The cumulative incidences of GI bleeding were similar by other factors in Table 2.

Furthermore, Table 3 presents the results of interaction between trial drugs and those factors (age, gender, race, aspirin use, and smoking status) on the risk of GI bleeding, but did not find any of these interactions significant. Table 4 classified the cumulative incidence at 3 different time intervals: at 1 year, 3 year and 5 year for overall population and also for the stratified results by age, sex, race, use of aspirin and estrogen, and smoking status. Although the cumulative incidence rates of GI bleeding increased substantially at 5 years as compared to those at 1 year, the cumulative incidence rates of GI bleeding were generally similar among the 3 trial arms (lisinopril, amlodipine, and chlorthalidone). However, in those with prior aspirin use and smoking group, patients with lisinopril had slightly higher incidence of GI bleeding (13.3% and 12.5% respectively) than those with amlodipine (10.2% and 9.8% respectively).

Table 5 presents the adjusted hazard ratios of GI bleeding for comparisons among 3 different trial arms as well as comparison among other factors. For example, as compared to those receiving chlorthalidone, patients who received amlodipine did not have a significantly different risk of developing GI bleeding [HR = 1.03, 95% CI: 0.87–1.22 from inpatient data, or 1.01 (0.90–1.14) from outpatient data]. The increased risk of GI bleeding by age was statistically significant after adjusting for confounders in the Cox regression models (HR = 1.04 per year

**Table 3. Hazard ratios (95% CI) for GI bleeding within subgroup by 3 RCT arms comparison.**

| | Hospitalized GI Bleeding | | Non-hospitalized GI Bleeding | | Hospitalized or non-hospitalized GI Bleeding | |
|---|---|---|---|---|---|---|
| | HR (95% CI) | *P value* | HR (95% CI) | *P value* | HR (95% CI) | *P value* |
| **Follow-up by 3/31/2002 events/total** | | | | | | |
| Chlorthalidone vs Amlodipine | 0.98 (0.83–1.15) | 0.78 | 1.07 (0.96–1.19) | 0.21 | 1.05 (0.95–1.16) | 0.35 |
| Lisinopril vs Amlodipine | 0.98 (0.82–1.16) | 0.79 | 1.01 (0.90–1.14) | 0.88 | 1.01 (0.91–1.13) | 0.81 |
| Lisinopril vs Chlorthalidone | 1.00 (0.85–1.16) | 0.95 | 0.95 (0.85–1.05) | 0.30 | 0.97 (0.88–1.07) | 0.53 |
| **Stratified by subgroups:** | | | | | | |
| In Black patients | | | | | | |
| Chlorthalidone vs Amlodipine | 0.93 (0.72–1.20) | 0.58 (0.49)* | 0.98 (0.83–1.17) | 0.86 (0.81)* | 0.97 (0.82–1.13) | 0.67 (0.37)* |
| Lisinopril vs Amlodipine | 1.10 (0.83–1.45) | 0.50 (0.90)* | 0.97 (0.80–1.18) | 0.76 (0.49)* | 1.02 (0.86–1.22) | 0.80 (0.57)* |
| Lisinopril vs Chlorthalidone | 1.18 (0.92–1.51) | 0.20 (0.40)* | 0.98 (0.83–1.17) | 0.85 (0.59)* | 1.06 (0.90–1.24) | 0.50 (0.79)* |
| In non-Black patients | | | | | | |
| Chlorthalidone vs Amlodipine | 1.05 (0.85–1.28) | 0.67 | 1.01 (0.88–1.16) | 0.86 | 1.06 (0.93–1.20) | 0.36 |
| Lisinopril vs Amlodipine | 1.07 (0.86–1.35) | 0.53 | 1.06 (0.91–1.23) | 0.47 | 1.09 (0.95–1.26) | 0.22 |
| Lisinopril vs Chlorthalidone | 1.03 (0.84–1.25) | 0.79 | 1.04 (0.91–1.19) | 0.53 | 1.03 (0.91–1.16) | 0.66 |
| Women | | | | | | |
| Chlorthalidone vs Amlodipine | 1.08 (0.88–1.34) | 0.47 | 1.00 (0.87–1.15) | 0.97 | 1.04 (0.91–1.18) | 0.58 |
| Lisinopril vs Amlodipine | 1.15 (0.91–1.45) | 0.25 | 1.01 (0.86–1.18) | 0.91 | 1.07 (0.92–1.24) | 0.36 |
| Lisinopril vs Chlorthalidone | 1.06 (0.86–1.30) | 0.57 | 1.01 (0.88–1.17) | 0.87 | 1.03 (0.91–1.17) | 0.64 |
| Men | | | | | | |
| Chlorthalidone vs Amlodipine | 0.90 (0.71–1.15) | 0.39 (0.26)* | 1.01 (0.86–1.19) | 0.92 (0.92)* | 1.01 (0.86–1.17) | 0.94 (0.76)* |
| Lisinopril vs Amlodipine | 1.00 (0.77–1.30) | 0.98 (0.46)* | 1.04 (0.87–1.25) | 0.66 (0.79)* | 1.05 (0.89–1.25) | 0.53 (0.90)* |
| Lisinopril vs Chlorthalidone | 1.11 (0.88–1.42) | 0.37 (0.76)* | 1.03 (0.88–1.21) | 0.70 (0.85)* | 1.05 (0.90–1.22) | 0.53 (0.86)* |
| Aspirin use at baseline | | | | | | |
| Chlorthalidone vs Amlodipine | 1.05 (0.79–1.38) | 0.75 (0.64)* | 0.99 (0.82–1.19) | 0.92 (0.87)* | 1.06 (0.89–1.25) | 0.53 (0.65)* |
| Lisinopril vs Amlodipine | 1.20 (0.89–1.62) | 0.23 (0.37)* | 1.18 (0.97–1.44) | 0.10 (0.09)* | 1.23 (1.02–1.48) | 0.03 (0.06)* |
| Lisinopril vs Chlorthalidone | 1.15 (0.88–1.49) | 0.30 (0.61)* | 1.19 (1.00–1.42) | 0.05 (0.04)* | 1.16 (0.99–1.36) | 0.07 (0.10)* |
| No-Aspirin use at baseline | | | | | | |
| Chlorthalidone vs Amlodipine | 0.97 (0.79–1.17) | 0.72 | 1.01 (0.89–1.15) | 0.89 | 1.01 (0.89–1.14) | 0.91 |
| Lisinopril vs Amlodipine | 1.02 (0.82–1.26) | 0.89 | 0.95 (0.82–1.11) | 0.55 | 0.99 (0.86–1.13) | 0.86 |
| Lisinopril vs Chlorthalidone | 1.05 (0.86–1.28) | 0.61 | 0.94 (0.83–1.08) | 0.40 | 0.98 (0.86–1.11) | 0.73 |
| Age 55–64 yrs | | | | | | |
| Chlorthalidone vs Amlodipine | 1.36 (0.77–2.42) | 0.29 (0.53)** | 0.84 (0.58–1.21) | 0.35 (0.35)** | 0.99 (0.70–1.39) | 0.94 (0.61)** |
| Lisinopril vs Amlodipine | 1.31 (0.69–2.50) | 0.42 (0.83)** | 0.99 (0.66–1.49) | 0.97 (0.95)** | 1.04 (0.71–1.53) | 0.83 (0.90)** |
| Lisinopril vs Chlorthalidone | 0.95 (0.55–1.65) | 0.87 (0.90)** | 1.17 (0.80–1.70) | 0.41 (0.55)** | 1.05 (0.74–1.48) | 0.78 (0.88)** |
| Age 65–69 yrs | | | | | | |
| Chlorthalidone vs Amlodipine | 0.95 (0.69–1.29) | 0.73 | 1.11 (0.92–1.34) | 0.29 | 1.10 (0.92–1.32) | 0.28 |
| Lisinopril vs Amlodipine | 1.05 (0.75–1.48) | 0.78 | 1.05 (0.85–1.30) | 0.63 | 1.11 (0.91–1.35) | 0.32 |
| Lisinopril vs Chlorthalidone | 1.11 (0.82–1.52) | 0.50 | 0.95 (0.79–1.14) | 0.59 | 1.00 (0.84–1.19) | 0.99 |
| Age 70 yrs or older | | | | | | |
| Chlorthalidone vs Amlodipine | 0.98 (0.81–1.19) | 0.87 | 0.97 (0.85–1.11) | 0.68 | 0.99 (0.87–1.12) | 0.86 |
| Lisinopril vs Amlodipine | 1.07 (0.86–1.32) | 0.55 | 1.01 (0.87–1.18) | 0.86 | 1.05 (0.91–1.20) | 0.54 |
| Lisinopril vs Chlorthalidone | 1.09 (0.90–1.31) | 0.40 | 1.04 (0.91–1.20) | 0.54 | 1.06 (0.93–1.20) | 0.39 |
| Non-smokers (ever/never) | | | | | | |
| Chlorthalidone vs Amlodipine | 1.00 (0.84–1.19) | 0.96 | 1.00 (0.89–1.12) | 0.97 | 1.02 (0.92–1.14) | 0.72 |
| Lisinopril vs Amlodipine | 1.07 (0.88–1.30) | 0.48 | 1.01 (0.88–1.14) | 0.93 | 1.05 (0.93–1.19) | 0.40 |
| Lisinopril vs Chlorthalidone | 1.08 (0.91–1.28) | 0.39 | 1.00 (0.89–1.13) | 0.96 | 1.03 (0.93–1.15) | 0.56 |

*(Continued)*

**Table 3.** (Continued)

| | Hospitalized GI Bleeding | | Non-hospitalized GI Bleeding | | Hospitalized or non-hospitalized GI Bleeding | |
| --- | --- | --- | --- | --- | --- | --- |
| | HR (95% CI) | *P value* | HR (95% CI) | *P value* | HR (95% CI) | *P value* |
| Smokers (current) | | | | | | |
| Chlorthalidone vs Amlodipine | 1.02 (0.70–1.49) | 0.91 (0.90)* | 1.00 (0.76–1.32) | 0.98 (0.97)* | 1.04 (0.81–1.34) | 0.75 (0.88)* |
| Lisinopril vs Amlodipine | 1.13 (0.75–1.71) | 0.56 (0.82)* | 1.12 (0.83–1.53) | 0.45 (0.51)* | 1.13 (0.85–1.49) | 0.41 (0.67)* |
| Lisinopril vs Chlorthalidone | 1.11 (0.76–1.60) | 0.59 (0.90)* | 1.13 (0.86–1.48) | 0.37 (0.43)* | 1.08 (0.84–1.39) | 0.54 (0.74)* |

* P value for interaction

**Likelihood ratio test P value for interaction.

increase, 95% CI: 1.03–1.05). Smokers also had a significantly higher risk of having hospitalized GI bleeding than those who did not (1.45, 1.19–1.76). Hispanics, those who used aspirin or atenolol at baseline, had diabetes, more education, and a history of MI or stroke had a significantly lower risk of having hospitalized GI bleeding than their counterparts, while only those who had diabetes had a significantly lower risk of having both hospitalized and non-hospitalized GI bleeding. Other factors such as gender, history of CHD, prior treatment of hypertension, use of estrogen in women, and obesity did not have significant effects on the risk of GI bleeding (Table 5).

## Discussion

This study demonstrated that the risk of hospitalized GI bleeding was not significantly different among the ALLHAT trial arms (amlodipine, lisinopril, and chlorthalidone) during the entire in-trial follow-up period that ended by March 31, 2002. These findings were consistent with previous ALLHAT reports which were based on a shorter term of follow-up data from Medicare [20–22]. Moreover, our new analyses on the risk of non-hospitalized GI bleeding and combined all GI bleeding (hospitalized or non-hospitalized GI bleeding) also showed no statistically significant differences among the 3 trial arms. However, subgroups with older age and with smokers were found to be at a higher risk for GI bleeding, whereas patients who used aspirin or atenolol prior to the trial, Hispanics, those with diabetes, more education, and a history of MI or stroke had a significantly lower risk of having GI bleeding than their counterparts. The risk of GI bleeding did not significantly vary by gender, history of CHD, prior treatment of hypertension, use of estrogen in women, and obesity.

We were able to have the data of ALLHAT trial participants linked with their Medicare claims data for the entire in-trial follow-up period through the Center for Medicare and Medicaid Services (CMS). Hence, our study was the only ALLHAT report with a complete follow-up information on the risk of GI bleeding during the entire in-trial period through March 31, 2002 and thus was able to capture all potential hospitalized GI bleeding events. All previous ALLHAT reports on GI bleeding were either based on a short follow-up data of Medicare data before [20, 22] or by September 24, 2001 [22]. Furthermore, all previous ALLHAT studies on GI bleeding were identified from inpatient hospitalization data only, and none of them had information on the GI bleeding identified from outpatient clinics or physician office visits. Our study was able to investigate relatively less severe GI bleeding events that did not lead to hospitalizations but were identified and treated in outpatient clinics or physician offices. Therefore, our report not only had complete follow-up information within the entire in-trial period but also reported for the first time the risk of GI bleeding from the outpatient data as

**Table 4. Cumulative proportion of participants with GI bleeding by year, subgroup, and RCT arms.**

| | Cumulative Incidence % (95% CI) of GI Bleeding per 1000 participants at the End of the Specified Year | | | | | |
| --- | --- | --- | --- | --- | --- | --- |
| | Hospitalized GI Bleeding | | | Hospitalized and Non-hospitalized GI Bleeding | | |
| | Year 1 | Year 3 | Year 5 | Year 1 | Year 3 | Year 5 |
| **Total** | | | | | | |
| Chlorthalidone | 6.4 (4.8–8.5) | 24.8 (21.6–28.5) | 47.0 (42.4–52.1) | 6.5 (5.0–8.6) | 50.4 (45.7–55.5) | 118.0 (110.8–125.7) |
| Amlodipine | 7.1 (5.0–10.0) | 24.7 (20.6–29.6) | 46.9 (41.0–53.6) | 7.5 (5.4–10.5) | 48.5 (42.6–55.1) | 111.3 (102.3–121.1) |
| Lisinopril | 9.4 (6.9–12.7) | 27.7 (23.3–32.9) | 50.0 (43.8–56.9) | 10.0 (7.5–13.4) | 51.6 (45.5–58.4) | 120.1 (110.7–130.3) |
| **Black** | | | | | | |
| Chlorthalidone | 6.7 (4.2–10.6) | 25.2 (20.0–31.9) | 50.1 (42.3–59.2) | 6.7 (4.2–10.6) | 46.8 (39.4–55.4) | 121.4 (109.4–134.7) |
| Amlodipine | 5.0 (2.5–9.9) | 30.3 (23.0–39.9) | 52.0 (42.1–64.3) | 5.0 (2.5–9.9) | 58.8 (48.4–71.4) | 121.3 (106.1–138.6) |
| Lisinopril | 9.6 (5.8–15.9) | 30.8 (23.3–40.6) | 53.8 (43.6–66.4) | 9.6 (5.8–15.9) | 50.6 (40.8–62.7) | 124.6 (108.8–142.5) |
| **Non-black** | | | | | | |
| Chlorthalidone | 6.2 (4.4–8.9) | 24.6 (20.6–29.3) | 45.4 (39.7–51.8) | 6.4 (4.6–9.1) | 52.4 (46.5–58.9) | 116.1 (107.3–125.7) |
| Amlodipine | 8.2 (5.5–12.2) | 21.6 (16.9–27.5) | 44.1 (37.1–52.5) | 8.9 (6.1–13.0) | 42.8 (36.0–50.8) | 105.8 (94.8–117.9) |
| Lisinopril | 9.2 (6.4–13.5) | 26.0 (20.8–32.5) | 47.9 (40.6–56.6) | 10.3 (7.2–14.7) | 52.1 (44.6–60.7) | 117.7 (106.2–130.3) |
| **Women** | | | | | | |
| Chlorthalidone | 7.3 (5.2–10.3) | 25.1 (20.9–30.2) | 48.5 (42.4–55.6) | 7.5 (5.4–10.6) | 49.7 (43.7–56.6) | 117.6 (108.1–127.8) |
| Amlodipine | 8.5 (5.6–12.9) | 22.9 (17.8–29.4) | 44.1 (36.7–53.1) | 8.9 (5.9–13.4) | 46.9 (39.4–55.8) | 111.4 (99.5–124.5) |
| Lisinopril | 9.6 (6.5–14.4) | 27.7 (22.0–35.0) | 50.9 (42.7–60.6) | 10.4 (7.1–15.3) | 49.8 (41.9–59.1) | 121.1 (108.5–135.1) |
| **Men** | | | | | | |
| Chlorthalidone | 5.2 (3.2–8.3) | 24.4 (19.7–30.3) | 44.9 (38.2–52.8) | 5.2 (3.2–8.3) | 51.3 (44.2–59.4) | 118.6 (107.7–130.5) |
| Amlodipine | 5.1 (2.8–9.5) | 27.1 (20.8–35.3) | 50.6 (41.6–61.5) | 5.6 (3.1–10.1) | 50.6 (41.8–61.3) | 111.2 (97.8–126.4) |
| Lisinopril | 9.0 (5.7–14.3) | 27.6 (21.3–35.8) | 48.8 (40.1–59.4) | 9.5 (6.1–14.9) | 53.7 (44.7–64.6) | 119.0 (105.3–134.4) |
| **Aspirin use** | | | | | | |
| Chlorthalidone | 8.2 (5.4–12.4) | 25.6 (20.3–32.3) | 44.9 (37.5–53.7) | 8.5 (5.7–12.8) | 51.2 (43.5–60.2) | 112.9 (101.2–125.9) |
| Amlodipine | 5.6 (2.9–10.8) | 20.1 (14.2–28.3) | 43.2 (34.1–54.7) | 5.6 (2.9–10.8) | 39.5 (31.0–50.3) | 102.3 (88.1–118.6) |
| Lisinopril | 8.0 (4.7–13.7) | 29.6 (22.4–39.0) | 48.6 (39.0–60.4) | 8.6 (5.1–14.5) | 56.7 (46.4–69.0) | 132.9 (117.0–150.8) |
| **No Aspirin use** | | | | | | |
| Chlorthalidone | 4.9 (3.3–7.4) | 24.1 (20.2–28.9) | 48.0 (42.2–54.6) | 4.9 (3.3–7.4) | 49.5 (43.7–56.0) | 120.9 (111.8–130.7) |
| Amlodipine | 8.0 (5.3–12.0) | 27.4 (22.0–34.0) | 49.5 (42.0–58.2) | 8.7 (5.9–12.8) | 53.0 (45.4–61.8) | 116.2 (104.7–128.8) |
| Lisinopril | 10.4 (7.2–14.9) | 26.5 (21.1–33.1) | 50.8 (43.1–59.9) | 11.1 (7.8–15.7) | 48.3 (41.0–56.9) | 113.4 (101.8–126.2) |
| **Age 55–64 yrs** | | | | | | |
| Chlorthalidone | 7.6 (3.2–18.2) | 32.0 (21.0–48.7) | 54.2 (39.2–74.7) | 7.6 (3.2–18.2) | 48.8 (34.7–68.3) | 116.9 (94.1–144.8) |
| Amlodipine | 0.0 (*—*) | 14.7 (6.6–32.4) | 36.8 (22.3–60.2) | 0.0 (*—*) | 53.9 (35.8–80.7) | 113.6 (85.6–150.2) |
| Lisinopril | 5.3 (1.3–21.2) | 16.0 (7.2–35.4) | 45.3 (27.7–73.4) | 8.0 (2.6–24.7) | 45.5 (28.5–72.1) | 129.3 (98.2–169.3) |
| **Age 65–69 yrs** | | | | | | |
| Chlorthalidone | 3.0 (1.5–6.0) | 15.4 (11.4–20.9) | 32.7 (26.4–40.4) | 3.0 (1.5–6.0) | 38.8 (32.1–46.8) | 103.3 (92.1–115.9) |
| Amlodipine | 3.8 (1.7–8.5) | 19.1 (13.4–27.2) | 37.6 (29.1–48.6) | 3.8 (1.7–8.5) | 38.8 (30.3–49.6) | 98.2 (84.1–114.6) |
| Lisinopril | 5.8 (3.0–11.1) | 23.9 (17.4–32.8) | 37.6 (29.0–48.6) | 6.5 (3.5–12.0) | 47.8 (38.2–59.6) | 103.9 (89.4–120.7) |
| **Age 70 or older** | | | | | | |
| Chlorthalidone | 8.3 (6.0–11.5) | 29.4 (24.8–34.9) | 54.6 (48.0–62.0) | 8.5 (6.2–11.7) | 57.7 (51.2–65.1) | 127.0 (117.2–137.6) |
| Amlodipine | 10.2 (6.9–14.9) | 29.7 (23.8–37.1) | 54.1 (45.8–63.9) | 10.9 (7.6–15.8) | 53.6 (45.5–63.0) | 119.0 (106.8–132.5) |
| Lisinopril | 12.1 (8.5–17.2) | 31.7 (25.6–39.2) | 58.2 (49.6–68.2) | 12.5 (8.9–17.7) | 54.8 (46.6–64.3) | 128.6 (115.8–142.6) |
| **Non-smokers (former/never)** | | | | | | |
| Chlorthalidone | 6.3 (4.6–8.5) | 24.0 (20.5–28.1) | 45.8 (40.8–51.4) | 6.4 (4.7–8.7) | 50.0 (44.9–55.7) | 118.0 (110.1–126.4) |
| Amlodipine | 7.4 (5.1–10.8) | 25.0 (20.4–30.5) | 46.8 (40.3–54.2) | 8.0 (5.6–11.4) | 48.1 (41.7–55.4) | 113.8 (103.9–124.7) |
| Lisinopril | 10.4 (7.6–14.2) | 26.6 (21.9–32.3) | 49.4 (42.8–57.0) | 11.2 (8.3–15.1) | 50.6 (44.0–58.1) | 119.2 (109.0–130.3) |
| **Smokers (current)** | | | | | | |

*(Continued)*

**Table 4.** (Continued)

| | Cumulative Incidence % (95% CI) of GI Bleeding per 1000 participants at the End of the Specified Year | | | | | |
| --- | --- | --- | --- | --- | --- | --- |
| | Hospitalized GI Bleeding | | | Hospitalized and Non-hospitalized GI Bleeding | | |
| | Year 1 | Year 3 | Year 5 | Year 1 | Year 3 | Year 5 |
| Chlorthalidone | 7.0 (3.7–13.4) | 28.8 (21.0–39.6) | 53.0 (41.6–67.3) | 7.0 (3.7–13.4) | 52.2 (41.3–65.9) | 118.1 (101.0–137.7) |
| Amlodipine | 5.2 (1.9–13.7) | 23.3 (14.8–36.8) | 47.8 (34.4–66.1) | 5.2 (1.9–13.7) | 50.5 (37.2–68.5) | 98.3 (78.8–122.3) |
| Lisinopril | 4.2 (1.3–12.8) | 33.2 (22.4–49.2) | 53.1 (38.7–72.6) | 4.2 (1.3–12.8) | 56.8 (42.1–76.3) | 125.1 (102.3–152.5) |

*confidence interval could not be calculated

well as the risk of GI bleeding among ALLHAT trial participants from the combined information from the inpatient and outpatient data. Moreover, our report corrected a few potential errors associated with the format of ICD-9 diagnosis coding and electronic codes in CMS datasets such as '05310' versus '5310' or '53100', which might lead to a small misclassification bias

**Table 5. The effect of antihypertensive treatment on the risk of hospitalized GI bleeding, non-hospitalized GI bleeding, and combined GI bleeding adjusting for baseline characteristics and in-trial use of aspirin and atenolol.**

| Covariates | HR* (95% CI) of Hospitalized GI Bleeding | HR* (95% CI) of Non-hospitalized GI Bleeding | HR* (95% CI) of Hospitalized or non-hospitalized GI Bleeding |
| --- | --- | --- | --- |
| Amlodipine vs Chlorthalidone | 1.03 (0.87–1.22) | 1.01 (0.90–1.14) | 0.99 (0.88–1.10) |
| Amlodipine vs Lisinopril | 1.00 (0.82–1.21) | 0.97 (0.85–1.10) | 0.95 (0.84–1.07) |
| Chlorthalidone vs Lisinopril | 0.97 (0.81–1.15) | 0.95 (0.85–1.07) | 0.96 (0.86–1.07) |
| Baseline aspirin use (yes vs no) | 1.07 (0.90–1.27) | 1.01 (0.90–1.14) | 1.06 (0.96–1.19) |
| Aspirin ever use prior to GI bleeding (yes vs no) | 0.67 (0.55–0.80) | 0.81 (0.72–0.92) | 0.75 (0.67–0.84) |
| Baseline Atenolol (yes vs no) | 0.68 (0.41–1.10) | 0.78 (0.58–1.06) | 0.78 (0.59–1.03) |
| Atenolol ever use prior to GI bleeding (yes vs no) | 0.76 (0.62–0.93) | 0.91 (0.80–1.03) | 0.86 (0.76–0.97) |
| Age (per year) | 1.04 (1.03–1.05) | 1.02 (1.01–1.03) | 1.02 (1.01–1.03) |
| Black (yes vs no) | 0.97 (0.82–1.15) | 1.11 (0.99–1.24) | 1.08 (0.97–1.19) |
| Sex (male vs women) | 1.05 (0.90–1.22) | 1.10 (0.99–1.22) | 1.07 (0.97–1.18) |
| Hispanics (yes vs no) | 0.70 (0.56–0.87) | 1.17 (1.03–1.34) | 1.04 (0.92–1.19) |
| Smoker (yes vs no) | 1.45 (1.19–1.76) | 1.01 (0.87–1.16) | 1.07 (0.94–1.22) |
| Education (per year) | 0.96 (0.95–0.98) | 0.99 (0.98–1.00) | 0.99 (0.97–1.00) |
| Diabetes (yes vs no) | 0.64 (0.54–0.74) | 0.88 (0.79–0.97) | 0.83 (0.76–0.92) |
| History of CHD (yes vs no) | 0.93 (0.76–1.15) | 1.02 (0.89–1.18) | 1.01 (0.88–1.15) |
| Atherosclerotic CVD (yes vs no) | 0.95 (0.75–1.21) | 0.96 (0.82–1.13) | 0.97 (0.83–1.12) |
| History MI or stroke (yes vs no) | 0.77 (0.63–0.95) | 0.93 (0.81–1.08) | 0.88 (0.77–1.01) |
| Coronary revascularization (yes vs no) | 0.93 (0.73–1.19) | 0.84 (0.71–0.99) | 0.86 (0.74–1.01) |
| Other atherosclerotic CVD (yes vs no) | 0.87 (0.71–1.06) | 0.92 (0.80–1.05) | 0.89 (0.78–1.01) |
| SBP (per mmHg) | 1.00 (1.00–1.01) | 1.00 (1.00–1.01) | 1.00 (1.00–1.00) |
| DBP (per mmHg) | 0.99 (0.98–1.00) | 1.00 (0.99–1.00) | 1.00 (0.99–1.00) |
| Antihypertensive (treated vs no) | 1.29 (0.97–1.72) | 1.13 (0.94–1.36) | 1.17 (0.99–1.40) |
| Women taking estrogen (yes vs no) | 0.77 (0.56–1.07) | 1.14 (0.95–1.37) | 1.06 (0.89–1.27) |
| BMI (per BMI score) | 0.99 (0.97–1.01) | 0.99 (0.98–1.01) | 0.99 (0.98–1.00) |
| Obesity (yes vs no) | 1.04 (0.82–1.31) | 1.03 (0.88–1.20) | 1.05 (0.90–1.21) |

*HR (hazard ratio) was adjusted for other variables in the Table.

on the study outcomes (see S1 Text). Therefore, our study findings added unique and insightful information to the existing literature on the findings of ALLHAT trial and should have important clinical and public health implications for the safety and side-effects of antihypertensive drugs, even though the study's follow-up time was date back to March 2002.

An early prospective community-based cohort study of 1636 hypertensive persons aged 68 years or older found that the use of calcium antagonists was associated with an increased risk of gastrointestinal hemorrhage with a relative risk of 1.86 (95% CI: 1.22–2.82) when compared with beta-blockers [11]. The similar finding was reported from a case-control study in Italy in 1998, in which the relative risk of GI bleeding associated with the current use of calcium channel blockers was 1.7 (95% CI: 1.3–2.1) [12]. A case-control in 1992–1994 from hypertensive patients from a Health Maintenance Organization group found that calcium channel blocker use was associated with a higher risk of lower GI tract bleeding, upper GI tract bleeding, and peptic ulcer-related bleeding [13]. Although three additional case-control studies also found a significant association between CCB use and the increased risk of GI bleeding [23–25], 6 other case-control studies did not find a significant association [15–17, 26–28]. Furthermore, a nested case-control study within a population-based cohort of all 34,074 found no significant association between CCB and GI bleeding [16]. A retrospective cohort study among 105,824 enrollees of the Tennessee Medicaid program 65 years of age or older between 1984 and 1986 also found no increased risk for hospitalization with bleeding peptic ulcer among users of calcium channel blockers [17]. Previous ALLHAT reports [20–22] and our study using the latest complete information on the risk of GI bleeding had similar findings and confirmed that CCB was not associated with an increased risk of both hospitalized and non-hospitalized GI bleeding as compared to patients receiving diuretics or ACE-inhibitors. Three other trials also found no significant association between CCB and the risk of GI bleeding [29–31]. A meta-analysis of 17 studies showed a marginal association between CCB and an increased risk of GI bleeding [32], but this association was largely driven by the above 5 case-control studies [12, 13, 23–25] and a cohort study [11], whereas 6 other case-control studies [15–17, 26–28], 1 cohort study [18], and 4 clinical trials [22, 29–31] did not find a significant association between CCB and the risk of GI bleeding.

This study has several limitations. First, our report did not have long-term follow-up information on the risk of GI bleeding for Canadian and VA participants in ALLHAT because of lack of their Medicare claims data. Second, the ALLHAT trial participants who had GI bleeding regardless of its severity but did not go to the outpatient clinics or were not hospitalized were obviously not captured in this dataset, hence the study likely underestimated the risk of outcomes. Third, the study only addressed the risk of GI bleeding by the end of the in-trial period on March 31, 2002 that was about 18 years ago. Hence, the findings might have minimal impact on the current clinical practice. However, its assurance of the safety of calcium-channel blockers and the other two drugs (diuretics and ACE-inhibitors) in terms of GI bleeding may still be relevant to the clinical prescription of antihypertensive drugs for patients with hypertension. Further studies may be considered to have a longer follow-up by taking into consideration post-trial antihypertensive drug utilization and their changing drug patterns. Fourth, although we assumed that GI bleeding identified from outpatient clinics may be less severe than those identified from inpatient hospitalization data, it was not possible to determine its validity because Medicare claims data do not have details to differentiate them.

In conclusion, there were no statistically significant differences in the risk of hospitalized GI bleeding and combined all GI bleeding (hospitalized or non-hospitalized GI bleeding) among the 3 ALLHAT trial arms (amlodipine, lisinopril, and chlorthalidone) during the entire in-trial follow-up that ended by March 31, 2002. However, subgroups with older age and with smokers were found to be at a higher risk for GI bleeding, whereas patients with prior aspirin

or atenolol use in-trial, Hispanics, those with diabetes, more education, and a history of MI or stroke had a significantly lower risk of having GI bleeding than their counterparts. The findings should have clinical and public health implications on the safety and choices of these antihypertensive drugs by physicians and patients with hypertension.

## Supporting information

**S1 Table. ICD-9 and ICD-10 codes for GI bleeding.**
(PDF)

**S1 Text. Statistical programs and models.**
(PDF)

## Acknowledgments

We thank Charles Coton for his participation and data management and analytical support for the study.

## Author Contributions

**Conceptualization:** Xianglin L. Du, Lara M. Simpson, Barry R. Davis.

**Data curation:** Xianglin L. Du, Lara M. Simpson, Brian C. Tandy, Barry R. Davis.

**Formal analysis:** Xianglin L. Du, Lara M. Simpson, Brian C. Tandy, Barry R. Davis.

**Funding acquisition:** Xianglin L. Du, Barry R. Davis.

**Investigation:** Xianglin L. Du, Barry R. Davis.

**Methodology:** Xianglin L. Du, Barry R. Davis.

**Project administration:** Xianglin L. Du.

**Resources:** Xianglin L. Du.

**Supervision:** Xianglin L. Du, Barry R. Davis.

**Validation:** Xianglin L. Du, Barry R. Davis.

**Visualization:** Xianglin L. Du, Barry R. Davis.

**Writing – original draft:** Xianglin L. Du, Lara M. Simpson, Brian C. Tandy, Judith L. Bettencourt, Barry R. Davis.

**Writing – review & editing:** Xianglin L. Du, Lara M. Simpson, Brian C. Tandy, Judith L. Bettencourt, Barry R. Davis.

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
