## [Decision Letter · Decision Letter 0]

22 Jul 2021

PONE-D-21-20128

Risk of  hospitalized and non-hospitalized gastrointestinal bleeding in ALLHAT trial participants receiving diuretic, ACE-inhibitor, or calcium-channel blocker

PLOS ONE

Dear Dr. Xianglin L Du,

Thank you for submitting your manuscript to PLOS ONE. After careful consideration, we feel that it has merit but does not fully meet PLOS ONE’s publication criteria as it currently stands. Therefore, we invite you to submit a revised version of the manuscript that addresses the points raised during the review process.

Please respond to each of the points made by reviewer 1 below.  

We look forward to receiving your revised manuscript.

Kind regards,

James M Wright

Academic Editor

PLOS ONE

Journal Requirements:

Reviewers' comments:

Reviewer's Responses to Questions

**Comments to the Author**

1. Is the manuscript technically sound, and do the data support the conclusions?

Reviewer #1: Yes

Reviewer #2: Yes

2. Has the statistical analysis been performed appropriately and rigorously? 

Reviewer #1: Yes

Reviewer #2: I Don't Know

3. Have the authors made all data underlying the findings in their manuscript fully available?

Reviewer #1: Yes

Reviewer #2: Yes

4. Is the manuscript presented in an intelligible fashion and written in standard English?

Reviewer #1: Yes

Reviewer #2: Yes

5. Review Comments to the Author

Reviewer #1: This is a useful study that further adds to the analysis of the ALLHAT study, showing that there is no significant difference between the antihypertensive medications used in the 3 arms of the trial regarding the incidence of GI bleeding. Through the use linked ALLHAT participant Medicare claims data taken to the end of the study, the authors were able to conduct a data linkage analysis that supports the conclusion about GI bleeding and the conclusion that age and smoking is associated to increased risk of GI bleeding while other factors are associated with reduced risk of GI bleeding. While conclusions drawn from the ALLHAT trial and accompanying analyses will always be limited by the rapidly updating clinical evidence, this study is important in expanding our understanding of the results of the trial for better completeness and for reviews of the existing evidence.

Major concerns

1. In Data Availability, relevant to the Methods section, the authors state “We plan to share the statistical models and statistical programs that we sued to analyze these data upon request”. The statistical programs and the versions used for the Kaplan-Meier method and the Cox regression models should be provided at the very least in the Statistical analysis section found on page 7.

Minor concerns

1. Authors state that the smoking group are at increased risk of GI bleed, and this is supported in Table 5 but only for hospitalized GI bleeding, it should be specified in the Results page 10 second paragraph “Smokers also had a significantly higher risk of having GI bleeding than those who did not” should be “Smokers also had a significantly higher risk of having hospitalized GI bleeding than those who did not”. Page 2 Results paragraph “Smokers also had a significantly higher risk of having GI bleeding” should be “Smokers also had a significantly higher risk of having hospitalized GI bleeding”.

2. The generalized conclusion statements about the relevance of this study for its clinical significance should be more specific. Page 5 first paragraph “Hence, the findings of this study should have high public health and clinical significance in terms of overall side effects and routine intake of antihypertensive drugs” should be “Hence, the findings of this study should have high public health and clinical significance with respect to GI bleeding and routine intake of antihypertensive drugs”. Page 12 paragraph one, line 4 “health implications for the safety and side-effects of antihypertensive drugs,” should be “health implications for the safety and side effects of antihypertensive drugs regarding GI bleeding,”

3. Minor grammatical clarification on page 12, paragraph one, line 5 “study’s follow-up time was back to March 2002.” should be “study’s follow-up time was dated back to March 2002.”

Reviewer #2: This is a very clear and simple study with important conclusions. I am not competent to judge the statistical methods used, but the study description, patient characteristics and findings, including limitations are all presented clearly and completely.

It should be published without any revisions.

6. PLOS authors have the option to publish the peer review history of their article (what does this mean?). If published, this will include your full peer review and any attached files.

Reviewer #1: No

Reviewer #2: No

---

## [Author Response · Author response to Decision Letter 0]

20 Oct 2021

Responses to the reviewer’s comments on a point-by-point basis. 

Reviewer #1: This is a useful study that further adds to the analysis of the ALLHAT study, showing that there is no significant difference between the antihypertensive medications used in the 3 arms of the trial regarding the incidence of GI bleeding. Through the use linked ALLHAT participant Medicare claims data taken to the end of the study, the authors were able to conduct a data linkage analysis that supports the conclusion about GI bleeding and the conclusion that age and smoking is associated to increased risk of GI bleeding while other factors are associated with reduced risk of GI bleeding. While conclusions drawn from the ALLHAT trial and accompanying analyses will always be limited by the rapidly updating clinical evidence, this study is important in expanding our understanding of the results of the trial for better completeness and for reviews of the existing evidence.

Major concerns

1. In Data Availability, relevant to the Methods section, the authors state “We plan to share the statistical models and statistical programs that we sued to analyze these data upon request”. The statistical programs and the versions used for the Kaplan-Meier method and the Cox regression models should be provided at the very least in the Statistical analysis section found on page 7.

Response: As suggested, we have added our statistical programs for the Kaplan-Meier estimates on the cumulative incidence of GI bleeding and the Cox regressions on the hazard ratio of GI bleeding in the “Data Availability” section. 

Minor concerns

1. Authors state that the smoking group are at increased risk of GI bleed, and this is supported in Table 5 but only for hospitalized GI bleeding, it should be specified in the Results page 10 second paragraph “Smokers also had a significantly higher risk of having GI bleeding than those who did not” should be “Smokers also had a significantly higher risk of having hospitalized GI bleeding than those who did not”. Page 2 Results paragraph “Smokers also had a significantly higher risk of having GI bleeding” should be “Smokers also had a significantly higher risk of having hospitalized GI bleeding”.

Response: Thank you for helpful suggestions. We have made these two changes as suggested.

2. The generalized conclusion statements about the relevance of this study for its clinical significance should be more specific. Page 5 first paragraph “Hence, the findings of this study should have high public health and clinical significance in terms of overall side effects and routine intake of antihypertensive drugs” should be “Hence, the findings of this study should have high public health and clinical significance with respect to GI bleeding and routine intake of antihypertensive drugs”. Page 12 paragraph one, line 4 “health implications for the safety and side-effects of antihypertensive drugs,” should be “health implications for the safety and side effects of antihypertensive drugs regarding GI bleeding,”

Response: Thank you for helpful suggestion. We have made this change as suggested on page 5. 

3. Minor grammatical clarification on page 12, paragraph one, line 5 “study’s follow-up time was back to March 2002.” should be “study’s follow-up time was dated back to March 2002.”

Response: Thank you for helpful suggestion. We have made this change as suggested on page 12.

Reviewer #2: This is a very clear and simple study with important conclusions. I am not competent to judge the statistical methods used, but the study description, patient characteristics and findings, including limitations are all presented clearly and completely.

It should be published without any revisions.

Response: Thank you so much for positive comments!

---

## [Editor Report · Decision Letter 1]

3 Nov 2021

Risk of  hospitalized and non-hospitalized gastrointestinal bleeding in ALLHAT trial participants receiving diuretic, ACE-inhibitor, or calcium-channel blocker

PONE-D-21-20128R1

Dear Dr. Xianglin L. Du: 

We’re pleased to inform you that your manuscript has been judged scientifically suitable for publication and will be formally accepted for publication once it meets all outstanding technical requirements.

Kind regards,

James M Wright

Academic Editor

PLOS ONE
---

## [Editor Report · Acceptance letter]

8 Nov 2021

PONE-D-21-20128R1 

Risk of  hospitalized and non-hospitalized gastrointestinal bleeding in ALLHAT trial participants receiving diuretic, ACE-inhibitor, or calcium-channel blocker 

Dear Dr. Du:

I'm pleased to inform you that your manuscript has been deemed suitable for publication in PLOS ONE. Congratulations! Your manuscript is now with our production department. 

Kind regards, 

on behalf of

Professor James M Wright 

Academic Editor

PLOS ONE